# Evaluating Caregiver Risk: The Dementia Caregiver Interview Guide

Rhonda Feldman [1,2,*], Mary Chiu [1,2], Andrea Lawson [2] and Joel Sadavoy [1,2,3]

1    Cyril & Dorothy, Joel & Jill Reitman Centre for Alzheimer's Support and Training, Mount Sinai Hospital, Sinai Health, Toronto, ON M5T 3L9, Canada; chium@ontarioshores.ca (M.C.); joel.sadavoy@sinaihealth.ca (J.S.)
2    Department of Psychiatry, Mount Sinai Hospital, Sinai Health System, Toronto, ON M5G 1X5, Canada; angewest@aol.com
3    Department of Psychiatry, University of Toronto, Toronto, ON M5T 1R8, Canada
*    Correspondence: rhonda.feldman@sinaihealth.ca; Tel.: +416-586-4800 (ext. 3212)

**Abstract:** Objectives: Family and other informal caregivers of individuals with dementia can be at increased risk for a significant decline in wellbeing or their ability to continue to provide care. There is extensive literature on the multifactorial elements contributing to risk, but frontline practitioners may be uncertain how to apply their knowledge of risk to an assessment of individual caregivers during clinical encounters. We developed a new one-page guided interview tool (the Dementia Caregiver Interview Guide, or DCIG) to guide practitioners to: (1) systematically assess known factors associated with high caregiver risk in a clinical interview format and (2) concisely document their judgement regarding risk of decompensation arising from caregiver stress. This semi-structured interview format collects detailed information while promoting a collaborative communication process. This study evaluated the validity of risk-assessment using the DCIG. Methods: A convenience sample of 50 caregivers was recruited during routine intake at the Reitman Centre at Sinai Health in Toronto, Canada. Risk was assessed using both the DCIG and the Caregiver Risk Screen (CRS). Total scores on the two tools were compared to establish concurrent and discriminant validity for the DCIG. Results: The DCIG correlated positively with the CRS (Spearman's rho = 0.737; $p < 0.001$) and identified caregivers at risk at a moderate level of agreement with the CRS (Cohen's Kappa = 0.559). Conclusions: The DCIG allows clinicians to efficiently identify caregivers' level of risk for functional and emotional decline or decompensation in a client-centered, naturalistic manner.

**Keywords:** dementia; caregivers; risk factors

## 1. Introduction

There is strong evidence that informal caregivers of individuals with dementia are at higher risk for poor physical health and greater psychological distress compared to their age-matched peers [1–5]. Women and adult child caregivers are especially at risk for depression and anxiety [6]. Caregivers are challenged to adapt to a constantly changing landscape as they provide care to someone whose illness inevitably worsens over time. The ability to cope with caregiving stress is impacted by a complex interplay between the caregiver's underlying personality structure, the nature of the care recipient's illness, the history of their relationship, and the social context in which they live [7,8]. Some caregivers can be at increased or high risk for a significant decline in their own wellbeing and thus their ability to continue to provide care [9–12]. High risk does not necessarily imply or equate with crisis. Rather, high risk can indicate the degree to which a caregiver's personal and environmental hardships increase the probability of falling into crisis [13]. Risk is a dynamic construct, often varying over time with changes in modifiable factors. The value in identifying those caregivers who become "high risk" and defining those factors that are contributing to risk is that interventions can be tailored to address key modifiable factors.

Various methods have been developed to evaluate caregiver risk, including in-depth structured interviews [9,14], brief self-assessment screening tools [15–18], and questions measuring caregiver distress embedded in larger care recipient assessments (e.g., Neuropsychiatric Inventory Caregiver Distress Scale [19]; Healthy Aging Brain Care Monitor [20]; interRAI Home Care [21]). Upon reviewing these methods for use in routine clinical assessment of dementia caregivers, it was determined that a different kind of tool was needed which could incorporate many of the positive qualities of other measures. Ideally this tool would be seamlessly embedded in the clinical assessment, facilitate full clinical engagement, and combine the thoroughness of the longer semi-structured interviews with the ease of use and brevity of the shorter questionnaires. To this end, we developed the Dementia Caregiver Interview Guide (DCIG) at The Cyril & Dorothy, Joel & Jill Reitman Centre for Alzheimer Support and Training ("Reitman Centre") at Sinai Health, a University of Toronto Academic Health Sciences Centre in Toronto, Canada. As part of the Geriatric Psychiatry Program at the hospital, the Reitman Centre has developed an array of multidimensional approaches to provide evidence-based skills training and therapeutic support for family caregivers who are caring for individuals with dementia [22–25].

One key construct regularly assessed in determining caregiving risk is caregiver burden, and numerous models have been proposed to capture the complexity of this construct [10,26–28]. The development of the DCIG is consistent with the principles laid down in Pearlin et al.'s (1990) Stress Process Model (SPM) [28], which describes caregiver stress as the result of interrelated stressors within a socioeconomic context.

Primary stressors are defined as struggles arising directly from the responsibilities of caregiving, while secondary stressors are those that occur as a result of the primary stressors and may include challenges to other roles or responsibilities or internal psychological conflict regarding self-identity. The stress of caregiving may be modified by implementing coping strategies and external support.

The development of the DCIG tool was also consistent with the definition of "Caregiver Assessment" provided by The Family Caregiver Alliance consensus [13], which stated that Caregiver Assessment is a systematic process of gathering information that describes a caregiving situation and identifies the particular problems, needs, resources, and strengths of the family caregiver. Following this definition, the DCIG was designed so that it approaches issues from the caregiver's perspective and culture, focuses on what assistance the caregiver may need and the outcomes the family member wants for support, and seeks to maintain the caregiver's own health and well-being. The tool also provides the assessor with an interview protocol that facilitates engagement and guides inquiry into known risk factors.

This study evaluated the validity and effectiveness of the DCIG in classifying caregiver risk during routine use in a clinical intervention program for caregivers. We hypothesized that the DCIG guided interview method would be as effective as an existing brief caregiver risk screen (Caregiver Risk Screen [CRS] [9]) in identifying caregivers at risk while maintaining a naturalistic flow of assessment and engagement of the caregiver. We evaluated (1) the DCIG's concurrent validity through correlation with the CRS and (2) the DCIG's discriminant ability to classify caregivers as high risk compared to the CRS.

## 2. Methods

### 2.1. Participants

The research methods and analyses for the study were approved by the Research Ethics Board at Mount Sinai Hospital, Sinai Health in Toronto, Canada. The study used a convenience sample of caregivers who were presenting for their first assessment at the Reitman Centre. Caregivers were included as study participants if they were able to converse in English and were informal and unpaid caregivers for an individual with dementia. There were no other inclusion criteria. Data were collected from April 2014 to July 2015.

*2.2. Measures*

2.2.1. Dementia Caregiver Interview Guide (DCIG)

The DCIG records a qualitative description of the caregiver's concerns, as well as the interviewer's impression of the degree of risk for caregiver decompensation. Most DCIG items were generated by reviewing literature on caregiver risk, burden, stress, and burnout [5,8,9,29–33]. Other items were generated through the extensive experience of the clinical team at the Reitman Centre which has been engaged in routine caregiver assessment since 2009. These "red flags" of caregiver risk were summarized in a one-page format. The tool was reviewed and updated iteratively based on feedback from content experts regarding face validity and clinical utility, both at the Reitman Centre and externally at community service agencies that provide support to both individuals with dementia and their caregivers. The final iteration was in keeping with the recommendations of the Family Caregiver Alliance Guidelines [13] for caregiver assessment including: multidimensional, family-centred, specific to each caregiver, practical and collaborative, leads to measurable outcomes, culturally competent, flexible in application, and empowering of the caregiver.

The DCIG is organized into 6 categories identified as relevant to risk in the literature and described in Table 1. Many of the categories are further divided into sub-categories, and a sample of included items is described in Table 1. To facilitate clinical documentation the DCIG is constructed in a checklist format for the interviewer to record those elements of risk that are revealed spontaneously by the caregiver or endorsed on further questioning (See Supplementary Materials). For each of the categories and sub-categories, the interviewer rates how strongly that element has impacted the caregiver's coping (low, medium, or high impact). Combined, the impact ratings made for each category qualitatively inform the interviewer's overall clinical impression.

**Table 1.** Categories and sub-categories with examples of items in the Dementia Caregiver Interview Guide (DCIG).

| Category | Sub-Category | Examples |
|---|---|---|
| (1) **Length or intensity of time spent providing care** | | |
| (2) **Care-recipient (CR) Characteristics:** | | |
| **Physical Issues:** e.g., increased dependency, substance use, and illnesses unrelated to dementia | | |
| **Dementia-specific Issues:** e.g., behavioural and psychological symptoms of dementia (BPSDs) | | |
| (3) **Caregiver (CG) Characteristics:** | | |
| **Psychological Characteristics:** e.g., psychiatric history, current feelings of anxiety, depression, guilt, or anger, feelings of poor mastery or self-efficacy, a history of conflict in their relationship with the care recipient, substance use | | |
| **Barriers to Change:** intrapsychic barriers to change which could inteR.F.ere with adaptability e.g., stigma/shame, denial of need, pressure from cultural group | | |
| **Physical Issues:** e.g., chronic or acute physical illness, age, disrupted sleep | | |
| **Dementia-specific Issues:** e.g., poor knowledge of dementia and inaccurate beliefs about behaviours | | |
| (4) **Environment:** | | |
| **Support:** e.g., family discord, lack of respite, transportation problems, poor program availability | | |
| **Resources:** e.g., financial or legal issues, employment stressors | | |
| (5) **Barriers to Accessing Resources:** e.g., marginalizing and systemic barriers to accessing resources due to language, culture, or sexual orientation | | |
| (6) **Caregiver-identified Overload:** pre-set question asking how many days in the last week they felt they had more to do than they could handle or "stretched to the limit" | | |
| (7) **Clinical Impression:** assessor uses a 7-point descriptive scale to rate their overall impression of the caregiver's risk of significant decline in their wellbeing or the care they can provide | | |

All caregivers experience a variety of stressors, but not all of them react to these factors in such a way that puts them at higher risk for their own psychological and physical decline. The DCIG Overall Clinical Impression scale allows the clinician to record their sense of risk for a significant decline in the caregiver's wellbeing or in the care they can provide. The assessor rates their degree of concern on a 7-point scale with descriptive terms ranging from "no concern" to "extreme". Scores of 5 (i.e., "marked concern") or above on the Overall Clinical Impression score are considered high risk. The 7-point scale format for the

DCIG Overall Clinical Impression of risk was based on the 7-point Likert-type scale used in the Clinical Global Impressions Scale [34]. The final determination of risk on the DCIG is guided by the impact ratings on each of the tool's categories but is a clinical, rather than a quantitatively derived, evaluation. For example, a caregiver's responses may generate a high overall impression of risk based on many sources of concern rated at a medium level of intensity or one item of high concern.

### 2.2.2. The Caregiver Risk Screen (CRS)

Of the existing caregiver risk measures, the CRS was chosen as a comparison measure because it is brief, addresses psychosocial elements of caregiver risk, and has clear scoring methods. The CRS was designed by Guberman et al. [9] to be a screening tool as part of a larger assessment process using a semi-structured interview called the C.A.R.E. Tool, "a psychosocial assessment tool to be used by home care practitioners with family caregivers to help understand Caregivers Aspirations, Realities, and Expectations (C.A.R.E)" [18]. The CRS tool is designed for home care agencies to determine the level of urgency for intervention with the family of an individual with dementia in the community. The 12 questions of the CRS are administered in a standardized assessment method. Each of the 12 statements are read to the family caregiver, who rates their agreement with each item on a 4-point scale from 0 (totally disagree) to 3 (totally agree) for a possible maximum of 36. The author's scoring guidelines suggest that scores of 17 or greater are considered high risk, and scores of 23 or greater are very high risk.

The screening tool was validated by the authors [9] and found to have adequate external consistency (Pearson correlation coefficient = 0.83, $p < 0.005$) and internal consistency (Alpha = 0.88). The tool has been validated in practice settings [35], evaluating health disparity in the U.S. [36], and chronic disease self-management support programs in Australia [37]. A Spanish version of the CRS has been validated [38]. For the current investigation the main authors were contacted, and permission was granted to use the CRS for comparison purposes.

### 2.3. Procedure

Fifty-six caregiver subjects provided verbal consent to participate during their initial intake call and written consent on arrival for their intake assessment appointment at the Reitman Centre. Four Reitman Centre mental health clinicians with extensive experience completing caregiver assessments collected the data. For each participating caregiver, the clinicians conducted an assessment using the established protocol at the Reitman Centre, which includes the DCIG. They administered the additional 12 item CRS questionnaire at the end of the standard assessment if consent had been provided. All forms were completed during the interview with the caregiver. Retrospective completion was not permitted. Both the CRS and DCIG were maintained confidentially as part of the caregiver's medical record at the Reitman Centre.

### 2.4. Statistical Analysis

All statistics were generated using SPSS 23 (Chicago, IL, USA). Descriptive statistics were calculated for the whole sample. The DCIG measure as a whole does not have traditional Likert scales that sum to a total score. Rather, it has three-point, non-summative category descriptions whose items aid the clinician in determining an Overall Clinical Impression final score. As such, traditional assessments of internal consistency such as Cronbach's alpha are not appropriate. Instead, Spearman's correlations between the individual items of the DCIG and the final Overall Clinical Impression item were calculated in order to assess the relative contribution of each item to the clinician's overall impression of a caregiver's risk. To assess concurrent validity, Spearman's correlations between the DCIG Overall Clinical Impression item and the CRS total score were calculated. Using linear weighted Cohen's kappa, the agreement in classification of caregivers as high risk

between the DCIG's Overall Clinical Impression item and the CRS total score was evaluated. Statistical significance was set with a *p* value of 0.05.

### 2.5. Sample Size

In order to determine discriminant and concurrent validity, the sample size was set at a minimum of 50. This sample size is adequate, with power set at 0.8 and probability of 0.05, to detect a Kappa of 0.8 (N ≥ 48) and a Spearman's rho of *r* = 0.273 (N ≥ 50). A Kappa of 0.61 to 0.80 is considered "good" while 0.81 to 1.00 is "very good" [36], and effect sizes of ≥0.4 are considered the minimum threshold as evidence of concurrent validity [39].

## 3. Results

### 3.1. Participant Demographics

Of the 96 full intake assessments conducted at the Reitman Centre during the data collection period, 56 caregivers provided written consent to participate in the study (representing a 58% participation rate). Reasons given for declining to participate included feeling too overwhelmed, feeling that the consent form was too long to read, and not wishing to participate in research. Of the 56 assessments completed with the DCIG and CRS measures, there were two CRS total values and four DCIG Overall Clinical Impression values missing due to assessor error. These participants were removed from the dataset, leaving an N = 50 for analyses. Demographic data for the participants can be found in Table 2.

**Table 2.** Demographic data for all participants.

|  | N | Mean | Std. Deviation |
|---|---|---|---|
| Caregiver Age | 50 | 63.80 | 15.18 |
| Care Recipient Age | 50 | 78.70 | 18.13 |
|  | | Frequency | Valid Percent |
| Caregiver Gender | | | |
| Male | | 13 | 26.5% |
| Female | | 36 | 73.5% |
| Total | | 49 | 100% |
| Does Caregiver live with Care Recipient? | | | |
| Yes | | 36 | 75.0% |
| No | | 12 | 25.0% |
| Total | | 48 | 100% |
| Care Recipient Gender | | | |
| Male | | 23 | 47.9% |
| Female | | 25 | 52.1% |
| Total | | 48 | 100% |
| Caregiver's Relationship with Care Recipient | | | |
| Spouse | | 26 | 53.1% |
| Child | | 22 | 44.1% |
| Sibling | | 0 | 2.0% |
| Other | | 1 | 0.0% |
| Total | | 50 | 100% |

### 3.2. Internal Consistency

Responses on the DCIG Overall Clinical Impression item were converted from the descriptive terms to a 7-point numeric scale (maximum score = 7). Spearman's correlations between the individual items of the DCIG and the final Overall Clinical Impression item were calculated in order to assess the relative contribution of each item to the clinician's overall impression of a caregiver's risk (see Table 3). The categories that most strongly correlated with the Overall Clinical Impression item were Caregiver Psychological Issues (Spearman's rho = 0.75, *p* < 0.001), Environmental Supports (Spearman's rho = 0.403,

$p = 0.004$), Environmental Resources (Spearman's rho = 0.62, $p < 0.001$), and Caregiver Self-Identified Overload (Spearman's rho = 0.66, $p < 0.001$).

**Table 3.** Spearman rank-order correlations of DCIG items with the Overall Clinical Impression.

| | **Spearman's Rho** | **p-Value** |
|---|---|---|
| Time spent in care | 0.39 | 0.007 |
| Care-recipient characteristics | | |
| Physical | 0.36 | 0.010 |
| Dementia-specific | 0.41 | 0.003 |
| Caregiver characteristics | | |
| Psychological | 0.75 | 0.000 |
| Barriers to change | 0.23 | 0.106 |
| Physical | 0.30 | 0.035 |
| Dementia-specific | 0.27 | 0.072 |
| Environment | | |
| Support | 0.40 | 0.004 |
| Resources | 0.62 | 0.000 |
| Barriers to accessing resources | 0.36 | 0.012 |
| Caregiver identified overload | 0.66 | 0.000 |

### 3.3. Concurrent Validity

The Spearman's rho between the CRS total score and the DCIG Overall Clinical Impression variables was 0.737 ($p < 0.001$). A correlation of this degree suggests that the DCIG and CRS are measuring the same underlying construct without being redundant.

### 3.4. Discriminant Validity

Using a cut-off of 5 ("marked concern") or above on the Overall Clinical Impression score of the DCIG as a determination of a caregiver as high risk, the rate of concordance with the CRS total score is 68.0% (n = 34). This represents fair agreement between the DCIG and the CRS (weighted Cohen's Kappa = 0.349, 95% CI: 0.11, 0.59) (see Table 4). Concordance improved when high risk was defined as 4 ("moderate concern") or above to 78% (n = 39), representing moderate agreement (weighted Cohen's Kappa = 0.559, 95% CI: 0.33, 0.79) (see Table 5).

**Table 4.** Two-by-two table of the DCIG overall clinical impression item with a cutoff for high risk of five or higher versus the CRS total score.

| **DCIG Overall Clinical Impression Item** | **CRS Total Score** | | |
|---|---|---|---|
| | **Not High Risk n( %)** | **High Risk n (%)** | **Total n (%)** |
| Not high risk | 23(46.0) | 13 (26.0) | 36 (72.0) |
| High risk | 3 (6.0) | 11 (22.0) | 14 (28.0) |
| Total | 26 (52.0) | 24 (48.0) | 50 (100.0) |

**Table 5.** Two-by-two table of the DCIG overall clinical impression item with a cut-off for high risk of four or higher versus the CRS total score.

| **DCIG Overall Clinical Impression Item** | **CRS Total Score** | | |
|---|---|---|---|
| | **Not High Risk n (%)** | **High Risk n (%)** | **Total n (%)** |
| Not high risk | 21(42.0) | 6 (12.0) | 27 (54.0) |
| High risk | 5 (10.0) | 18 (36.0) | 23 (46.0) |
| Total | 26 (52.0) | 24 (48.0) | 50 (100.0) |

## 4. Discussion

The Dementia Caregiver Interview Guide (DCIG) was developed with the objective of systematizing the evaluation of a caregiver's needs during routine clinical assessment, to highlight the most pertinent issues, contribute to a clinical case formulation, and help define areas for intervention. Tool creation was based on factors identified in the literature and clinical practice as potentially important to caregiver functioning and risk of decline. Study results provided evidence that the DCIG is strongly correlated with an existing validated measure of risk [9], capturing the same construct while identifying unique elements of risk.

Our findings show "moderate" agreement between the measures in their identification of caregivers as high risk. We acknowledge that this level of agreement is not definitive, given that the confidence intervals around the kappa value spans a wide range. This may reflect the inherently subjective nature of the interpretation of what constitutes "high risk" in a caregiver on the DCIG. The current results may also reflect the difference between a descriptive tool and a quantitative measure, as the CRS is a self-assessment rating scale with a forced-choice response format, while the DCIG is an embedded descriptive tool reflecting the clinician's synthesis of all relevant factors of risk as expressed by the caregiver. As such, the DCIG also differs from other existing measures of caregiver risk. Similar to the CRS, other tools [15–17] also use a questionnaire format, either asking caregivers to provide ratings of agreement or yes/no responses to items.

Self-assessment questionnaires are well-suited to capture caregivers' subjective perspective. However, some of our participants objected to the CRS forced-choice questionnaire format, as they felt unable to use discreet rating values to describe their experience. In some ways, using a forced-choice questionnaire imposes on the clinician a pre-specified wording of inquiry, and similarly imposes a pre-specified wording of the caregiver's response. As such, this type of tool can inteR.F.ere with the organic communication and connection between clinician and caregiver. An advantage of the DCIG is its ability to promote clinical engagement while simultaneously allowing evaluation. Our results showed that the clinician's Overall Clinical Impression score correlated strongly with the Caregiver Self-Identified Overload question, indicating that the DCIG successfully captured the caregiver's perspective.

*Limitations and Future Directions*

One must consider limits to the scope and generalization of a clinician-dependent evaluation tool. The DCIG incorporates many of the recommended elements for caregiver assessment [13,40], but as Zarit [8] notes, assessment must be tailored to the environment in which it is used and no one assessment tool can address all elements. The DCIG can complement an existing caregiver evaluation protocol as a way to document qualitative elements which may not be captured by scales with a forced choice questionnaire format.

The Reitman Centre's clinicians are highly skilled mental health professionals and are knowledgeable about the complexity of caregiver issues. As such, their ratings of Overall Clinical Impression could reflect rater bias, stressing mental health issues over other concerns. Indeed, the sub-category of Caregiver Psychological Issues strongly correlated with the Overall Clinical Impression score. Although the components of the DCIG can prompt interviewers regarding known caregiver risk factors, the tool could have different properties if used by assessors who may be less familiar with caregiver concerns or are in other clinical contexts. Thus, future research will need to evaluate inter-rater reliability within and between professional centres. Although there has been evidence that clinical recommendations vary by discipline [41], case formulations require clinical experience and judgement. The DCIG documents the degree of clinical concern about a caregiver's coping, which is the first step needed for case formulations and service recommendations.

Since this study was completed, the DCIG has been adopted by numerous clinicians who assess caregiver needs for intervention and support programs. Future practical considerations related to DCIG include an exploration of its clinical utility, including whether shorter versions of the tool can fit more fast-paced environments, such as primary

care doctor's practices, while maintaining its validity. Future iterations of the tool will also include an evaluation of positive protective factors that moderate sources of caregiver stress. With modifications to the scoring methods and follow-up of assessed caregivers over time, the DCIG may be employed to examine which factors of caregiver and care recipient experience may be associated with or predict degrees of risk and strength in caregiving

## 5. Conclusions

This validation study suggests that the Dementia Caregiver Interview Guide (DCIG) is able to identify caregivers at high risk for significant decline in their own wellbeing or care via an embedded semi-structured interview approach when compared to the CRS questionnaire. The one-page tool guides clinicians in their interview with caregivers using a summary of factors associated with caregiver risk. The DCIG provides a concise, valid process to elicit, evaluate, and document the degree of impact of these factors and concern about the caregiver's risk of emotional and physical decline.

**Supplementary Materials:** The following are available online at https://www.mdpi.com/article/10.3390/psych3040036/s1, File S1: Dementia Caregiver Interview Guide.

**Author Contributions:** Conceptualization, R.F. and J.S.; methodology, R.F., M.C., and J.S.; software, M.C.; validation, R.F., M.C. and J.S.; formal analysis, A.L.; investigation, R.F.; resources, J.S.; data curation, M.C.; writing—original draft preparation, R.F.; writing—review and editing, R.F., J.S., M.C., and A.L.; visualization, R.F.; supervision, J.S.; project administration, J.S.; funding acquisition, J.S. All authors have read and agreed to the published version of the manuscript.

**Funding:** This work was supported by the Behavioural Supports Ontario and Behavioural Supports for Seniors Program of the Toronto Central LHIN.

**Institutional Review Board Statement:** The study was conducted according to the guidelines of the Declaration of Helsinki, and approved by the Institutional Review Board of Sinai Health.

**Informed Consent Statement:** Informed consent was obtained from all subjects involved in the study.

**Data Availability Statement:** The data presented in this study are available in Feldman, R.; Chiu, M.; Lawson, A.; Sadavoy, J. Evaluating Caregiver Risk: The Dementia Caregiver Interview Guide.

**Acknowledgments:** The authors wish to thank Jennifer Carr, Sarah Gillespie, Gita Lakhanpal, and Dunstan Pushpakumar for their contributions to this project.

**Conflicts of Interest:** The authors report no conflicts of interest.

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
