# Peer review of "Evaluating Caregiver Risk: The Dementia Caregiver Interview Guide"

_psych, doi:10.3390/psych3040036_

Round 1

Reviewer 1 Report

Overall, a very interesting and valuable paper in a field that is increasingly focussing on caregiver wellbeing. The DCIG tool provides a more appropriate, qualitative analysis of the perspectives of those caring for dementia patients when compared with the CRS (and other known tools). 

The introduction provides an excellent background to the current issues faced in dementia care and the gaps that are regularly seen in terms of care givers. I note that many of the sources cited are quite old, although some of these are foundation references.  

There is strength in the statistical methods utilised to validate this tool, however I would be interested to know why a factor analysis/PCA was not utilised? Also, the data collected is from 7+ years ago - are there any further data to support the use of the DCIG since this work was done? Will there be a follow up study? It is mentioned in the limitations, but not clear as to what further studies will be produced in terms of use and outcomes of the DCIG.  

The results are clearly written, however the formatting of the tables makes them difficult to read. I would suggest re-formatting Table 1 and Table 3 and ensuring the headings are clear within your tables. Table 4 is very well presented.  It is great to see some correlation between the 2 tools and also well described reasoning for this correlation. In particular, the limitations around the moderate agreement described in your discussion and limitations section show high level thinking and respect for use of these type of tools. 

The discussion and conclusions sum up your findings nicely  and make a valuable contribution to the current body of knowledge and tools utilised in dementia care. Although, again it would be beneficial to include some more recent references. Well done. 

Author Response

Reviewer 1

Thank you for these helpful comments. In our response we have explained our reasoning and made changes in the paper where required.

Point 1: “I note that many of the sources cited are quite old, although some of these are foundation references”.  

Response: We agree. Updated references have been added but we have left in some of the foundation references which we believe are still relevant despite the passage of time.

Point 2: There is strength in the statistical methods utilised to validate this tool, however I would be interested to know why a factor analysis/PCA was not utilised?

Response: Re: factor analysis. The DCIG does not rely on summative scores for each item. Rather it is an interview guide to assist clinicians in systematic inquiry into caregiver needs and vulnerability. The inquiry is meant to allow clinicians to reveal these areas of vulnerability with patients and clients and hopefully determine points of intervention. The purpose of the study was to determine the overall validity and effectiveness of the tool in classifying overall level of caregiver risk when compared to another “gold standard” self-assessment tool. Hence, data was not gathered which would permit factor analysis of each item since this was not a goal of the study. We did apply Spearman’s correlations between the individual items of the DCIG and the final overall Clinical Impression item to assess the relative contribution of each item to the clinician’s overall impression of a caregiver’s risk. The items themselves were developed from an a priori analysis of research that revealed factors contributing to caregiver risk. 

Point 3: Are there any further data to support the use of the DCIG since this work was done? Will there be a follow up study? It is mentioned in the limitations, but not clear as to what further studies will be produced in terms of use and outcomes of the DCIG”. 

Response: Regrettably no further data is available yet. The reviewer's comment was helpful in highlighting this issue and to help frame future research questions although evaluation of these factors in the present study was beyond the purpose and scope of this study design. We added an update sentence  in the paper on the extensive clinical use being made of the tool which may offer opportunities to extend its use and research potential in this population depending on resources. We added additional clarification in the “limitations and  future directions section” indicating that future research may use a modified DCIG to determine  which  factors are  most relevant to understanding and determining caregiver risk.

Point 4: “The formatting of the tables makes them difficult to read. I would suggest re-formatting Table 1 and Table 3 and ensuring the headings are clear within your tables.

Response: It appears that formatting was lost in some tables when the paper was uploaded last time.  These tables have been reformatted.

Reviewer 2 Report

This is an interesting study that presents a useful tool for working with caregivers. I leave some suggestions for improvement in the text:

(a) 2.3 - Procedure: Caregivers' reactions depend on many factors, including cultural ones. Thus, it would be interesting to use a larger number of participants.

(b) Table 2  - The caregivers' reactions must be considered in a specific context, where the care situation is only one of the important dimensions. Previous physical and psychological problems in caregivers should also be analyzed.

(c)

Author Response

Reviewer 2

The reviewer noted 2 wise and helpful points for consideration, highlighting them in the text of the paper as well as in narrative.  We have explained our process and reasoning for each point raised.

Point 1: Re Procedure and sample size: “Caregivers' reactions depend on many factors, including cultural ones. Thus, it would be interesting to use a larger number of participants”.

Response:  Re: sample size: The reviewer cogently highlighted the complex factors that need to be considered in caregiver research. We considered these factors as we designed the study and looked carefully at the factors to be taken into account in determining the sample size for a study of this type i.e. one that that had the main purpose and focus of assessing the internal consistency of this tool and it’s concurrent and discriminant  validity  as compared to another “gold standard” self assessment tool. It was on this basis that we determined the sample size necessary to provide sufficient statistical power. For this particular study, caregiver profile per se did not appear to interfere with the statistical reliability of the study based on this sample size.  We tried to explain clearly in the paper the rational for sample size.

Point 2: Re  Table 2  - The caregivers' reactions must be considered in a specific context, where the care situation is only one of the important dimensions. Previous physical and psychological problems in caregivers should also be analyzed.

The main purpose of this study was to evaluate the DCIG rather than the individual participants. We fully agree with the reviewer‘s important observation that caregiver reactions must be considered within a specific context which takes into account both caregiver characteristics and the care situation. These factors are carefully considered in the sections of the DCIG itself, guiding assessors to inquire deeply about these matters with each caregiver. However, the study was not designed to assess the caregivers per se but rather the overall reliability of the tool in enabling assessors to evaluate overall risk. For this study, we did not attempt to define the demographic characteristics of the caregiver participants beyond those in the table although in other work listed in the references which looked at interventions for caregivers, we were very mindful of taking these factors into account. 

Round 2

Reviewer 2 Report

As I mentioned earlier, this is an interesting study for all who contribute to improving care for people with dementia and their family caregivers.

I found that the new version introduced some suggestions for improvement and I agree that other suggestions would be difficult to implement.

Thus, if this is the Editor's understanding, I consider that the article can be published.